# Oxidative Stress Levels Induced by Mercury Exposure in Amazon Juvenile Populations in Brazil

**DOI:** 10.3390/ijerph16152682

**Published:** 2019-07-27

**Authors:** Leandro V.B. Carvalho, Sandra S. Hacon, Claudia M. Vega, Jucilene A. Vieira, Ariane L. Larentis, Rita C. O. C. Mattos, Daniel Valente, Isabele C. Costa-Amaral, Dennys S. Mourão, Gabriela P. Silva, Beatriz F. A. Oliveira

**Affiliations:** 1Toxicology Laboratory, Center for Studies of Worker’s Health and Human Ecology (CESTEH), National School of Public Health Sergio Arouca (ENSP), Oswaldo Cruz Foundation (Fiocruz), Leopoldo Bulhões Street, 1480, Manguinhos, Rio de Janeiro-RJ, CEP 21041-210, Brazil; 2Department of Endemics Samuel Pessoa (DENSP), National School of Public Health Sergio Arouca (ENSP), Oswaldo Cruz Foundation (Fiocruz), Leopoldo Bulhões Street, 1480, Manguinhos, Rio de Janeiro-RJ, CEP 21041-210, Brazil; 3Centro de Innovaccion Cientifica Amazonica (CINCIA), Jr. Cajamarca c 1 s/n, Puerto Candamo, Puerto Maldonado, Madre de Dios, PERU - Wake Forest University, 1834 Wake Forest Rd, Winston-Salem, NC 27109, USA

**Keywords:** oxidative stress, mercury exposure, biomarkers, juvenile riverine communities

## Abstract

Oxidative stress can be induced by mercury (Hg) exposure, including through fish consumption (diet), leading to health risks. The objective of this study was to evaluate the association between oxidative stress biomarkers and dietary Hg exposure levels in riverine children and adolescents at Madeira River (RO/Brazil). Population from three riverine local communities presenting different fish consumption frequencies was sampled. Hg was determined in blood (ICP-MS) and glutathione (GSH); glutathione S-transferases (GST) and malondialdehyde (MDA) were determined in serum (spectrophotometry). Statistical analyses were performed using parametric and non-parametric tests. Multiple linear regression models and generalized additives models were also used to estimate the relationships between oxidative stress biomarkers and blood Hg. The juvenile riverine population from Cuniã RESEX presented the highest levels of oxidative stress and Hg levels in blood (GST = 27.2 (4.93) U/L, MDA = 1.69 (0.27) µmol/L, Hg = 20.6 (18.0) µg/L). This population also presented the highest frequency of fish consumption. The positive relation between Hg and GST and MDA, adjusted for individual characteristics, suggests an oxidative effect. This study shows the importance of oxidative stress biomarkers in the evaluation of dietary Hg exposure since initial and reversible metabolic changes were observed, enriching health risk assessments.

## 1. Introduction

Oxidative stress is an imbalance between the formation of reactive (pro-oxidant) species and the antioxidant defense system, leading to potential cellular damage [1,2]. The production of reactive species is part of the human metabolism and has important biological functions; as well as the normal aging process, however, imbalances are associated with diseases [2,3,4,5,6]. The evaluation of oxidative stress markers, using biological exposure indicators or biomarkers, is an important approach in environmental and/or occupational exposure studies and may provide early information on preclinical metabolic changes (non-observed health effects) related to disease progression [7,8,9,10,11].

The study of oxidative stress biomarkers induced by mercury (Hg) exposure is a way to sub clinically evaluate the effects of environmental exposure to this metal and its compounds [8,12,13] and may significantly alert to the health risks of the exposed population. In addition, they can be used to estimate risks in exposed populations, although they cannot be clinically used as disease indicator. The advantage of using this type of biomarker is that, once the source of exposure ceases, their levels can return to basal due to homeostasis, independent of the initial exposure level [7,14].

Hg is a neurotoxic element whose toxicity depends on the chemical species and exposure route. Methylmercury (MeHg), an organic form of Hg, can cross the blood-brain and placental barriers, which may cause irreversible damage during development. MeHg has properties of bioaccumulation and biomagnification, reaching high concentrations in top-chain organisms. Fish consumption is one the main means of human exposure to MeHg [15,16,17,18].

Studies have investigated the association between Hg exposure and oxidative stress in human adult populations that present high frequencies of fish consumption [13,19], but few studies discussing this relationship in children groups are available [20]. Due to their differentiated and easily alterable metabolism, children are a subgroup with specific vulnerabilities to toxic substances present in the environment [21,22]).

In the Amazon, metallic Hg has been used for long time to extract gold, but discussion of contamination problems and its effects in human health is scarce [23]. Riverine, or riparian, Amazon populations that depend on local fish for subsistence are at greater risk of developing health problems, mainly neurological issues, caused by MeHg [24]. Children are an extremely vulnerable group in these populations [25,26,27,28].

In this context, the aim of the present study was to evaluate oxidative stress levels in juvenile riverine populations, comprising children and adolescents between 5 and 17 years old at Madeira River (RO/Brazil). Malondialdehyde (MDA) and Glutathione (GSH) levels and Glutathione S-Transferase (GST) activity in serum were assessed as biomarkers of effect, and their association with Hg exposure, using Hg-Blood (Hg-B) as a biomarker for Hg, was evaluated.

## 2. Materials and Methods

### 2.1. Study Population

This study was developed in the Western Amazon region, at Madeira River (the main Amazon River tributary), located in the municipality of Porto Velho, capital of the state of Rondônia, Brazil.

Three local communities with different lifestyles and degrees of urbanization were selected for evaluation: An urban community in the city of Porto Velho; an urban riverside community (Belmont), located 12 km from the city; and an isolated riverside community (Cuniã Extractive Reserve-RESEX), located 180 km from the city. Belmont and Cuniã are fishing communities bordering the Madeira River. At Cuniã, the community also presents nut extractivism as one of their main subsistence activities. The urban Porto Velho community is located in the urban area of Porto Velho, also near the Madeira river, presenting a not very representative fish intake (Figure 1).

The selection of the three communities (Cuniã, Belmont and Porto Velho) was based on historic Hg exposure through fish consumption [30,31,32,33]. The Porto Velho urban community was selected as a reference population, due to low fish consumption.

### 2.2. Study Design and Population

A cross-sectional environmental exposure assessment study was carried out between 2012 and 2013. Blood samples were collected in August 2012, during the dry season (winter). The study selected children and adolescents between 5 and 17 years old as target groups from the three Amazonian communities (Figure 2).

The following inclusion criteria were adopted: Children and adolescents aged 5 to 17 residing in the study areas enrolled in local schools and local residents for at least 12 months who agreed to blood collection authorized by their parents. The exclusion criteria consisted in reports of any serious neurological disease (self-reported) or refusal to participate in the study at any time and, in the case of girls, being pregnant.

A participatory approach was developed where the project was explained to the community members, who were then invited to voluntarily participate in the study as well as in lectures, and to the staff of the schools of first and second grades. A semi-structured questionnaire (individual, one per participant) was applied, by interviewing the children’s parents, in order to obtain the sociodemographic characteristics and fish consumption frequency (biweekly, weekly—one to three times, or more than three times—and daily) of the participants.

This study was approved by the Research Ethics Committee of the National School of Public Health Sergio Arouca (ENSP)/Oswaldo Cruz Foundation (Fiocruz)—RJ/Brazil: CEP/ENSP 809.593, CAAE 18634613.0.0000.5240, in 8/22/2014. All participants had the consent of a parent or guardian, who signed a free and informed consent form.

### 2.3. Collection and Fractionation of Blood Samples

Whole blood samples were collected (*n* = 197) from the study participants using sterile syringes and transferred to Vacuette^®^ vacuum tubes. The tubes containing the anticoagulant sodium heparin (specific for trace metal analyses) were stored at −20 °C for subsequent determination of total Hg. Tubes without the anticoagulant were used to obtain serum for the oxidative stress biomarkers analyses. After the blood collection and transfer, the tubes were centrifuged (1600× *g* 10 min) and the serum was separated into different aliquots stored at −80 °C until the MDA, GSH and GST analyses. Blood samples were collected in August 2012, and samples (total blood and serum) were maintained frozen (−80 °C) until analysis, performed between August 2012 and October 2013.

### 2.4. Laboratory Analyses

All reagents were PA grade, supplied by Sigma-Aldrich/Vetec (St. Louis, MO, USA). Purified Type I water was used, obtained from a Merck Millipore purifying system (Darmstadt, Germany). The MDA analysis kit was supplied by the Cayman Chemical Company (Ann Arbor, MI, USA).

#### 2.4.1. Oxidative Stress Analyses

The oxidative stress analyses were all performed in serum, as this matrix correlates with the general effects of oxidative stress in the organism [34,35,36], by spectrophotometry on Jasco V-530 (Kyoto, Japan) and Shimadzu UV-1601 (Kyoto, Japan) UV-Vis spectrophotometers.

##### Determination of GST Activity

The determination of enzymatic GST activity followed the method described by Habig et al. (1974) [37], adapted by Habdous et al. (2002) [38]. The method is based on the reaction between 1-chloro-2,4-dinitrobenzene (CDNB) and reduced glutathione (GSH), catalyzed by GST. The reaction medium consists of 0.1 mol/L potassium phosphate buffer and pH 5.5 + 25 mmol/L CDNB + 50 mmol/L GSH. After adding the sample, the formation of the product was monitored for 5 min at 340 nm. The results were reported as units per liter (U/L).

##### Determination of GSH Levels

GSH concentrations were determined by the method described by Hu (1994) [39], since this method is related to the amount of total thiols in plasma. The reaction medium consists of 0.25 mol/L Tris-Cl Buffer with 0.02 mol/L EDTA + 0.01 mol/L dithiobis-nitrobenzoic acid (DTNB). An analytical curve was prepared using an external GSH standard. Absorbances were determined at 412 nm and the results were reported in milimol per liter (mmol/L). The limit of detection of the method was of 0.09 mmol/L (mM).

##### Determination of MDA Levels

MDA concentrations were determined using a commercial kit from the Cayman Chemical Company, based on the reaction of MDA with thiobarbituric acid (TBA) and sodium dodecyl sulfate (SDS) under boiling at 100 °C. Absorbances were determined at 532 nm, and the results were reported in micromol per liter (µmol/L). The limit of detection was of 0.79 µmol/L (µM).

#### 2.4.2. Determination of Hg Levels

The determination of the Hg biomarker in blood followed an adaptation of the methodology reported by Palmer et al. (2006) [40], with sample pre-treatment by acid digestion under heating at 80 °C. Hg determinations were performed using the Inductively Coupled Plasma-Mass Spectrometry (ICP-MS) technique on a PerkinElmer NexIon 300 (Waltham, MA, USA) equipment, and the results were reported in micrograms per liter (µg/L). For quality assurance and control, blanks were measured every day. The reference material, BIO RAD Lyphochek Whole Blood Metals Control, Level 2, was analyzed for accuracy, presenting recovery values above 85%.

### 2.5. Statistical Analyses

Descriptive statistics were performed to illustrate the demographic and socioeconomic characteristics of each community. Concentrations of oxidative stress biomarkers and Hg were expressed as means and standard deviation (SD). Both parametric and non-parametric tests according to variable distributions. For GST, GSH and Hg, data distribution was non-parametric (Kolmogorov–Smirnov Test with Lilliefors correction; *p*-value < 0.01), while for MDA the distribution was approximately parametric (Kolmogorov–Smirnov Test with Lilliefors correction; *p*-value = 0.040). Thus, for the non-parametric variables, the Mann–Whitney or Kruskall–Wallis (to compare two and two or more unpaired samples) were applied, while for parametric data the Unpaired T Test and ANOVA one-way were used (to compare two and two or more unpaired samples). In case of difference among the three groups, the Mann–Whitney test was applied for in between-groups comparisons with Bonferroni’s correction for non-parametric variables, while a One-way ANOVA with post-hoc Tukey HSD test was used for comparisons between the groups. In the descriptive analysis, the correlations between biomarkers were verified using Spearman’s test. Multiple Linear Regression Models and Generalized Additive Models were used to estimate the relationships between the oxidative stress biomarkers and blood Hg. The models were adjusted by age, gender and Body Mass Index (BMI). Linear regression analyses were performed by natural log (ln—transformation for non-parametric variables (GSH and GST). Natural cubic spline functions were used to capture potentials nonlinearities between Hg and the oxidative biomarkers. Model adjustments were carried out by a stepwise procedure, and their fit was assessed by an ANOVA test, the adjusted R2, and the Akaike Information Criteria (AIC). *p*-values < 0.05 were considered significant. All statistical analyses were performed using the R software package, version 3.3.1 (The R Foundation for Statistical Computing, Vienna, Austria, http://www.r-project.org) and the SPSS software package for social sciences (SPSS v. 20.0 for Windows, SPSS Inc., Chicago, IL, USA).

## 3. Results

A total of 197 children and adolescents were assessed, with mean age of 11.1 ± 2.7, comprising 60% females and 47% in the adolescence phase. No statistically significant differences were found between the demographic, social, BMI and residence time characteristics among the communities. Statistically significant percentage differences were observed among the three study sites for weekly fish consumption, with the highest (77%) for children and adolescents living in the Cuniã reserve, consuming fish over 3 times a week (Table 1).

The means and standard deviations of the oxidative stress biomarkers in children and adolescents for the entire group were of 0.48 (0.11) mmol/L for GSH, 19.8 (7.50) U/L for GST, and 1.45 (0.32) µmol/L for MDA. The results per group are described in Table 1, as well as Hg levels.

When comparing biomarker levels, statistically significant differences between the three communities for GST enzymatic activity were found (22.4 (7.77) U/L for Belmont, 27.2 (4.93) for Cuniã and 15.2 (4.42) for Porto Velho, *p*-value < 0.000), as well as differences for MDA between Cuniã and Belmont (1.69 (0.27) vs. 1.34 (0.28) µmol/L, *p*-value = 0.001) and Cuniã and Porto Velho (1.69 (0.27) vs. 1.37 (0.31) µmol/L, *p*-value < 0.000) (Table 1). Regarding Hg concentrations, differences between Cuniã and Belmont values (20.6 (18.0) vs. 7.84 (11.0) µg/L, *p*-value < 0.000) and Cuniã and Porto Velho (20.6 (18.0) vs. 5.22 (6.04) µg/L, *p*-value < 0.000) were observed (Table 1). For GSH, no differences were observed between the studied communities.

Hg exposure related to the frequency of fish consumption among the three communities was also compared (Table 1). The chi-square test (X^2^) confirmed (*p*-value < 0.001) that weekly fish consumption was statistically higher at Cuniã. A significant difference between the fish consumption profile at Belmont and Porto Velho was also observed (X^2^ test, *p*-value < 0.05), higher at Belmont.

The differences presented in Table 1 are shown as boxplots (Figure 3) for MDA, GSH, GST, and Hg-B. Oxidative stress biomarkers and Hg levels differences related to the frequency of fish consumption are also displayed.

Hg-B presented a moderate and significant correlation with GST (r = 0.39, *p*-value < 0.001) and a weak but significant correlation with MDA (r = 0.23, *p*-value < 0.001). MDA was moderately and significantly correlated to GST (r = 0.28, *p*-value < 0.001). Additionally, associations between body mass index (BMI) and age with oxidative stress and Hg-B were verified. BMI showed a weak and significant negative correlation with GST (r = −0.15, *p*-value < 0.05), while age was correlated to MDA (r = −0.16, *p*-value < 0.05). The associations between the assessed biomarkers and age, BMI and Hg-B are displayed in Table 2.

To understand the behavior of each biomarker associated to the Hg-B (log values) measured in blood, smoothed function graphs (Figure 4) were constructed, indicating the relationship between Hg-B levels and organism responses, i.e., metabolism and its association with effects due to oxidative stress.

## 4. Discussion

Oxidative stress biomarkers represent effects on the metabolism, interconnected with each other and with the illness process [2,41,42,43]. Understanding the role of biomarkers in human mercury exposure is a way to create alerts regarding the subclinical effects on human health and develop actions to mitigate and minimize the health risks of exposed populations.

GST enzymatic activity was significantly different among the three communities, with the highest activity observed at Cuniã. In addition, higher enzyme activity was associated to higher Hg-B levels, which, in turn, was associated to higher fish consumption. The mean GST activity in serum found in this study corroborates other reported in the literature. In a study carried out by Habdous et al. (2002) [38] on the determination of serum GST enzymatic activity, a value of 28.2 (3.0) U/L was suggested as a reference value for children and adults, while Kaynar et al. (2005) [44] observed mean GST activity of 16.8 (7.9) U/g Hb in erythrocytes of healthy adults.

GST is associated with the xenobiotic metabolism (phase II of detoxification processes). GST conjugates reduced glutathione (GSH) to an electrophilic substrate in the cell cytosol, facilitating excretion [45,46]. Other functions, such as the removal of reactive species and their products resulting from oxidative stress, are also associated with these enzymes [5,47,48]. In the context of this study, higher GST activity was expected in situations of higher Hg exposure.

The MDA results obtained in the present study were within the normal values reported in the literature, between 1–4 µmol/L [49,50,51,52], as well as near the reference value provided by the commercial kit used in the analysis, between 1.86–3.94 μmol/L. A statistically significant difference was observed between Cuniã and the other two communities, while Belmont and Porto Velho did not differ significantly between each other.

A positive association was observed between MDA and Hg-B. This demonstrates the importance of MDA as a biomarker of effect in Hg exposure assessments, as discussed by some authors [20,25,53], where increases in lipid peroxidation levels alongside higher Hg-B levels were observed.

The correlation observed between GST activity and Hg-B levels was higher than that noted between MDA and Hg-B, although both were positive and significant (Table 2). In addition, MDA and GST presented positive associations with each other. These results indicate that both biomarkers are associated with Hg-B levels, although GST was demonstrated as more sensitive regarding Hg exposure.

Figure 4 allows for a better visualization of the relationships between the assessed biomarkers. Graph A indicates that GSH and Hg-B values increased linearly. This may be because increasing Hg concentrations lead to increasing GSH (available sulfhydryl grouping), required to bind to Hg. Graph B indicates that GST activity displays different behaviors in relation to Hg-B concentrations. The activity of this enzyme may, at first, be enough for Hg metabolism. After a certain concentration, around 2.7 µg/L, GST must increase its activity to detoxify Hg, which may occur through positive feedback mechanisms [45,46]. From 12 µg/L, this route seems to be saturated, and GST activity is stabilized, no longer presenting significant increases. Graph C presents the relationship between MDA and Hg-B. Up to 5 µg/L of Hg, MDA concentrations do not increase significantly, probably because within this range of Hg concentration the redox equilibrium is maintained without generating oxidative damage. Thus, it seems that oxidative species are eliminated before generating damage, as in the case of lipid peroxidation, represented herein by the MDA product. From 5 µg/L Hg-B, MDA concentrations indicate a rapid and progressive increase of reactive species and, consequently, increasing damage to the exposed organism [2].

No differences were observed among the three communities regarding GSH levels. The mean serum GSH values were close to those reported by Costa et al. (2006) [54] for healthy patients (0.48 (0.18) mmol/L for men and 0.49 (0.17) mmol/L for women), although the age group evaluated by those authors was different from that assessed herein.

Glutathione (GSH), in the form of thiol, is more susceptible to oxidative damage since it is polarizable and acts as a nucleophile [39,47], displaying a direct relationship to Hg toxicity, where depletion of GSH levels is expected [55,56]. However, serum GSH levels in the present study showed no significant differences between the studied communities concerning to assess Hg exposure, although this does not exclude its importance as an oxidative stress biomarker related to this metal as demonstrated by other studies [12,13,20,57] where negative associations were found between glutathione levels and Hg. Another important factor to consider, which may have contributed to find no differences of GSH among the studied communities, is that this molecule undergoes self- oxidation very easily, depending on the conditions of collection and storage [58]. The GSH auto-oxidation of our samples, due to the difficulties of field work in a remote region, may have interfered in the results obtained, leading to this indifference of GSH levels between the groups mentioned.

The results observed herein for GSH may be due to the intake of high amounts of selenium (Se) present, in Brazil nuts [59,60], very common in riverine communities dietary. Selenol (~SeOH) are more nucleophilic than thiol groupings (~SH), which makes selenoproteins preferential molecular targets for MeHg compared to compounds containing protein thiols. This fact is based on the higher affinity of Hg for selenols when compared to thiols, which also makes Se-Hg binding stable even in the presence of high thiol concentrations [12,61]. Se levels in Cuniã have been previously assessed by Vega et al. (2017) [29] who reported that Se levels in blood are actually higher when compared to the Belmont community. This may be a possible hypothesis for the fact that no decreases in GSH levels in the population who is in fact most exposed to Hg (Cuniã) were observed.

The isolated Cuniã riverside community presented the Hg-B highest concentrations in relation to Belmont and Porto Velho. This exposure is related to fish consumption, obtained from Madeira River, displaying Hg contamination in its organic form, MeHg [30,31,32]. As the Cuniã RESEX is an isolated riverside community where fish is one of the main protein sources, this result was expected. These data confirm what certain authors have described in the literature, that riparian communities present higher Hg levels in comparison to urban populations [62,63].

These data were also reported by Hacon et al. (2014) [32] and Vega (2015) [33], in the same site of the present study. The authors observed that the highest Hg-B levels were found at Cuniã when compared to the other studied riparian and urban communities, due to the higher frequency of fish consumption, mainly carnivorous species, with higher Hg concentrations.

At Cuniã, the mean Hg levels were 2.5 times higher than the reference value of 8 µg/L, recommended by the World Health Organization (WHO) for non-exposed populations [16,18]. Regarding this community, 71% of the evaluated juveniles presented Hg-B levels above 8 µg/L. At Porto Velho and Belmont, on the other hand, only 14% and 31% of children, respectively, presented levels above 8 µg/L.

For the purpose of comparisons with populations from a different region, Xavier et al. (2013) [64] reported Hg-B concentrations in 220 school children (8–10 years old), of both sexes, from two schools belonging to the municipal school network in the metropolitan region of the city of Rio de Janeiro (Rio de Janeiro, RJ, Brazil). The mean value of Hg-B concentrations was of 0.89 µg/L, while the median was 0.71 µg/L, which is below that of the present study. However, that study was carried out in a site with no mercury contamination.

Riparian children and adolescents have a historical relationship with their residence site, since they grow consuming fish containing high Hg concentrations [24,31,32], which characterizes the riverine population as chronically exposed to the toxicological risks of Hg, with possible implications regarding cognitive development [17,28]. This reinforces the importance of studies evaluating exposure biomarkers that evidence health risks in very early stages.

The assessment of oxidative stress and its biomarkers is observed as a useful tool in the evaluation of exposure to chemical substances, since they give early information about the pathogenesis of a disease before the appearance of health effects [65].

An important discussion is the use of non-specific biomarkers, such as oxidative stress biomarkers, in exposure assessment studies. The more sensitive a biomarker, even if it is non-specific, the earlier an initial biochemical change can be detected, and still be reversible. This sensitivity gives them an important predictive capacity, but their alteration is multicausal, which makes it difficult to associate them with certain outcomes. Thus, the associated use of specific exposure biomarkers (internal dose and effect) for a certain compound in these types of studies can improve correlation levels. Thus, after careful analyses, associations between exposure and effect can be discussed with greater statistical validity and reliability.

## 5. Conclusions

This is the first study assessing oxidative stress biomarkers (GSH, MDA, and GST) and Hg exposure via fish intake at the Madeira River basin. The biomarkers Hg, MDA, and GST were found to be significantly higher at Cuniã when compared to the other evaluated communities. Only GSH levels showed no difference between the studied communities, indicating this is a non-sensitive biomarker in the context of this study. The riparian community at Cuniã presented the highest levels of dietary Hg exposure, due to higher frequency of fish consumption.

This study demonstrates the importance of oxidative stress biomarkers in the evaluation of dietary Hg exposure, since they indicate initial and reversible metabolic changes, enriching health risk assessments. The sensitivity of these biomarkers to low exposure levels is useful in alerting health authorities before a disease scenario develops, leading to more serious effects on human health.

## Figures and Tables

**Figure 1 ijerph-16-02682-f001:**
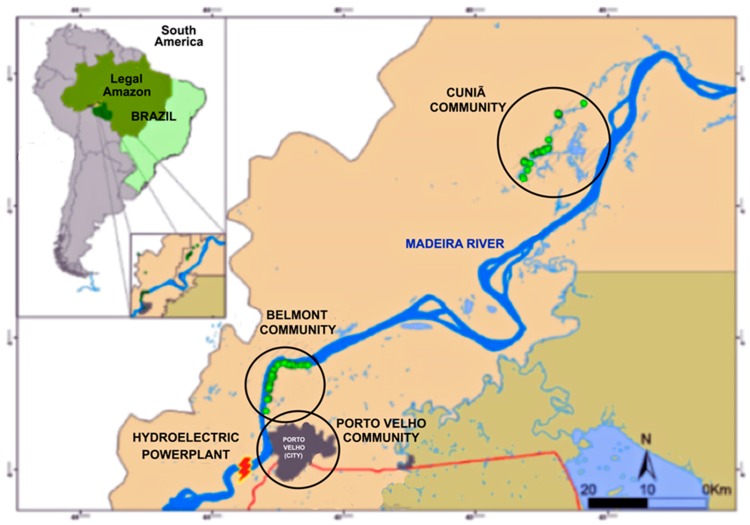
Map of the study area. Cuniã and Belmont are located downstream of Porto Velho city, in the state of Rondônia, Brazil. The circles represent the three communities (Belmont, Cuniã and Porto Velho) sampled in this study [29]. Reproduced with permission from Claudia M. Vega, Biological Trace Element Research; published by Springer Link, 2017.

**Figure 2 ijerph-16-02682-f002:**
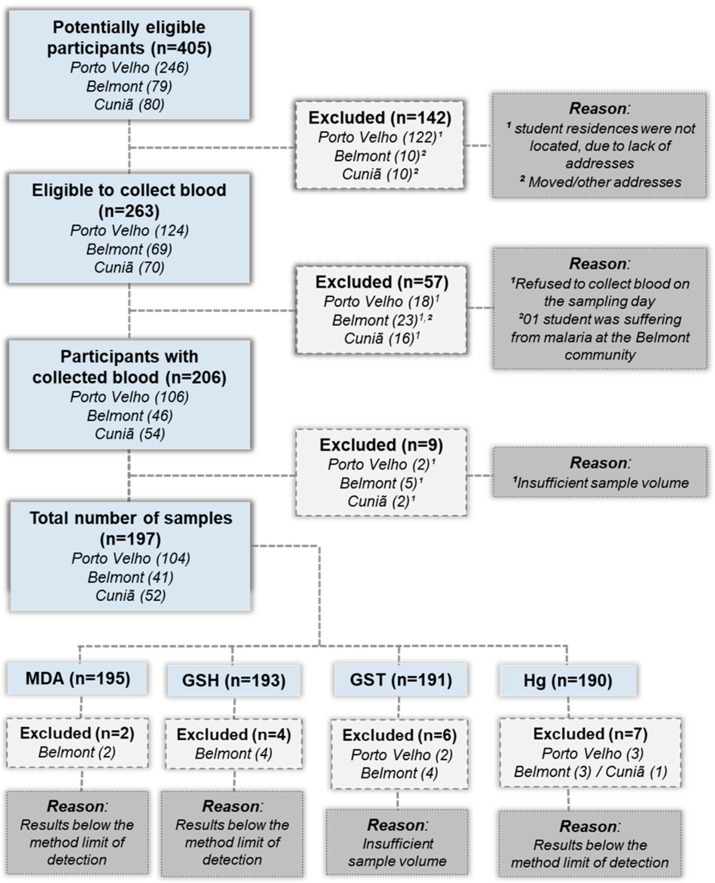
Flow diagram displaying the selection of the study population (children and adolescents, between 5 and 17 years old) in the Madeira basin, Rondônia/Brazil.

**Figure 3 ijerph-16-02682-f003:**
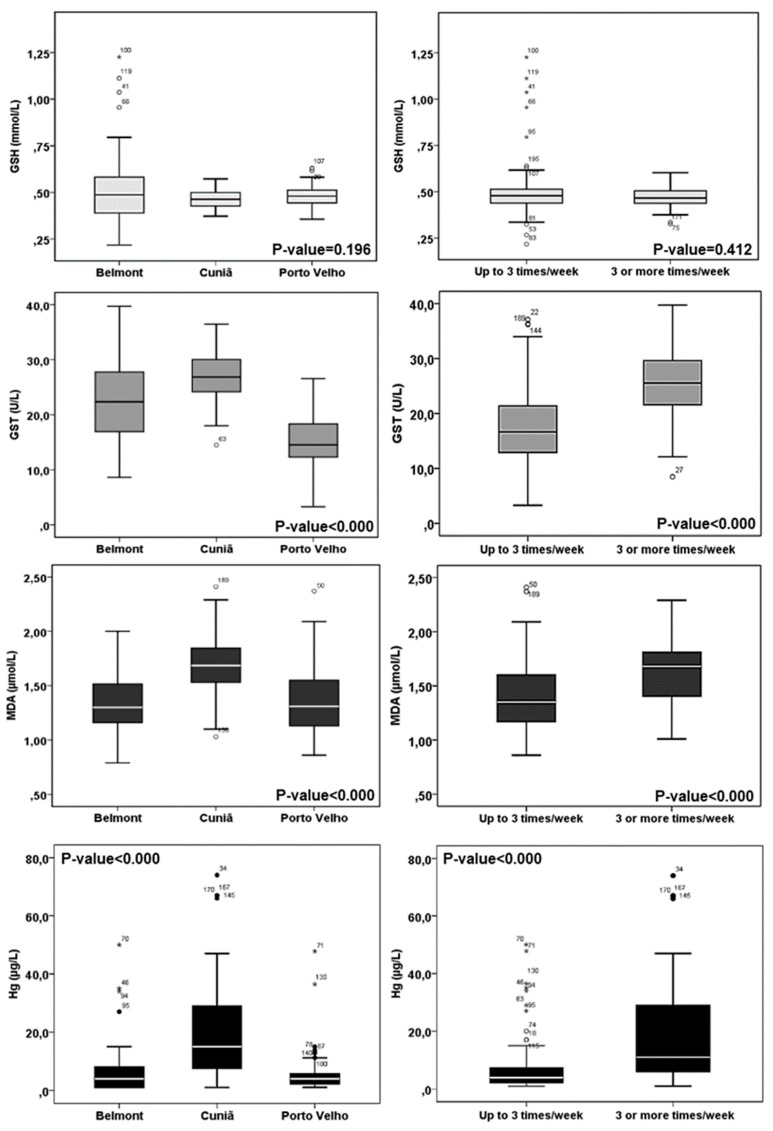
Boxplots displaying the results of the analyzed biomarkers (GSH, GST, MDA, and Hg), in the study population, per juvenile community and fish consumption frequency at the Madeira basin. Small dots outside the boxplots represent the outliers.

**Figure 4 ijerph-16-02682-f004:**
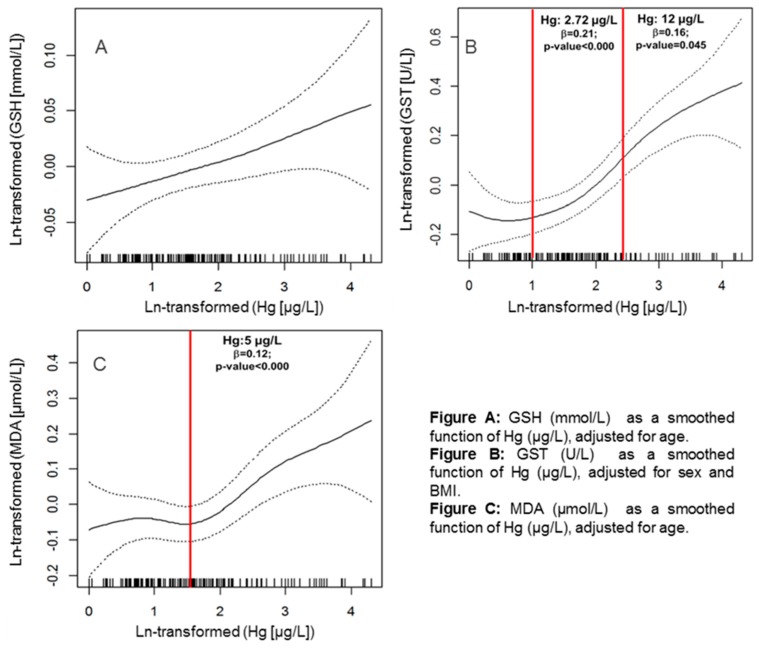
Smoothed function of oxidative stress biomarkers and their relationship with Hg. (**A**) GSH as a smoothed function of total mercury concentrations in blood (Hg-B), coefficient of ln (Hg): 0.024 (*p*-value = 0.052). (**B**) GST as a smoothed function of Hg-B. (**C**) MDA as a smoothed function of Hg-B. The relationship between GSH and Hg was linear. The relationship between Hg and GST was established using a spline with 2 knots inserted in the ln-transformed values of Hg (1 and 2.5 ≅ 2.72 and 12 µg/L of Hg). The relationship between Hg and MDA was established using a spline with df = 2.

**Table 1 ijerph-16-02682-t001:** Sociodemographic variables and biomarker results in the study population per juvenile community.

Communities	Belmont	Cuniã	Porto Velho	Total	*p*-Value
	*n* = 41	*n* = 52	*n* = 104	*n* = 197	
	N (%) or Means (SD)	N (%) or Means (SD)	N (%) or Means (SD)	N (%) or Means (SD)	
**Gender**					
Male	19 (46.3%)	22 (42.3%)	37 (35.6%)	78 (39.6%)	0.492
Female	22 (53.7%)	30 (57.7%)	67 (64.4%)	119 (60.4%)	
**Age (yrs)**	11.3 (3.07)	10.9 (2.51)	11.1 (2.61)	11.1 (2.68)	0.807
**Age Group**					
5–11 yrs	22 (53.7%)	30 (57.7%)	52 (50.0%)	104 (53.0%)	0.657
12–17 yrs	19 (46.3%)	22 (42.3%)	52 (50.0%)	93 (47.0%)	
**BMI**	18.5 (3.73)	17.5 (2.79)	18.4 (3.74)	18.2 (3.52)	0.455
**Residence Time (yrs)**	6.36 (2.42)	5.00 (1.56)	5.54 (2.34)	5.60 (2.25)	0.255
**Fish Consumption**					
0–3 times/week	33 (82.5%)	12 (23.1%)	100 (96.2%)	145 (74.0%)	0.000 ^1^
3 or + times/week	7 (17.5%)	40 (76.9%)	4 (3.8%)	51 (26.0%)	
**Oxidative Stress**					
GSH (mmol/L)	0.53 (0.22)	0.46 (0.04)	0.47 (0.05)	0.48 (0.11)	0.196
GST (U/L)	22.4 (7.77)	27.2 (4.93)	15.2 (4.42)	19.8 (7.50)	0.000 ^2^
MDA (µg/L)	1.34 (0.28)	1.69 (0.27)	1.37 (0.31)	1.45 (0.32)	0.000 ^3^
**Hg (µg/L)**	7.84 (11.0)	20.6 (18.0)	5.22 (6.04)	9.86 (13.1)	0.000 ^2^

^1^ Chi-Square test - Difference between Cuniã vs. Belmont vs. Porto Velho; ^2^ Kruskal–Wallis; and Mann–Whitney Test with Bonferroni correction for pairwise comparisons—Difference between Cuniã vs. Belmont vs. Porto Velho for GST (U/L) and difference between Cuniã vs. Belmont and Porto Velho for Hg (µg/L); ^3^ ANOVA One-Way Test; and post hoc Tukey HSD for multiple comparisons—Difference between Cuniã vs. Belmont and Porto Velho; BMI = body mass index.

**Table 2 ijerph-16-02682-t002:** Spearman correlations between the evaluated biomarkers, age, and BMI.

	GSH	GST	MDA	Hg	Age	BMI
**GSH**	1					
**GST**	0.128	1				
**MDA**	−0.048	**0.277 ****	1			
**Hg**	0.142	**0.388 ****	**0.229 ****	1		
**Age**	0.081	−0.113	**−0.160 ***	−0.072	1	
**BMI**	−0.064	**−0.152 ***	−0.051	−0.035	**0.489 ****	1

* Correlation is significant at the 0.05 level (2-tailed). ** Correlation is significant at the 0.01 level (2-tailed). Ln-transformed: GSH, GST, Hg, Age, and BMI.

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
