# Peer review of "Oxidative Stress Levels Induced by Mercury Exposure in Amazon Juvenile Populations in Brazil"

_ijerph, 2019, doi:10.3390/ijerph16152682_

Round 1

Reviewer 1 Report

The study is very interesting and the authors show the effect of mercury in some indicators of oxidative stress in children population. However, glutathione is a low molecular weight antioxidant that is very prone to oxidation when samples are stored and this can affect levels. Therefore it is suggested to the authors to explain how to stabilize the samples to avoid the oxidation of reduced glutathione. In addition, GST catalyze conjugation of reduced glutathione to a wide range of substrates, then how can the enzyme activity be increased without affecting the glutathione reduced. Please explain more deeply.

Author Response

Dear Reviewer

Thank you for the opportunity to review our manuscript. We very much appreciate the comments and suggestions from your revision, and maked the necessary adjustments in the manuscript.

All recommendations and questions were answered below (in red).

Thank you again for your time and consideration,

In case of any questions, please do not hesitate to contact us anytime.

Best wishes,

Leandro Vargas B. de Carvalho

Response to Reviewer 1 Comments

English language and style are fine/minor spell check required

Response: The manuscript was entirely revised by an english native speaker.

1) The study is very interesting and the authors show the effect of mercury in some indicators of oxidative stress in children population. However, glutathione is a low molecular weight antioxidant that is very prone to oxidation when samples are stored and this can affect levels. Therefore it is suggested to the authors to explain how to stabilize the samples to avoid the oxidation of reduced glutathione. In addition, GST catalyze conjugation of reduced glutathione to a wide range of substrates, then how can the enzyme activity be increased without affecting the glutathione reduced. Please explain more deeply.

1) R. Our method employed in the GSH analysis did not use any feature to stabilize the serum. However, the samples were storage in ultrafreezer (-80 ºC), to control, reduce and/or avoid the oxidation of the GSH.

Serum GSH analysis was one of several biomarkers to evaluate the levels of oxidative stress induced by Hg uptake via fish consumption.

For GSH analysis the classical method (Hu, 1994) was used, by adding the DTNB Ellman's reagent. Analyzes plasma/serum GSH, as well as all other plasma/serum thiol groups, also were included in our study. So, we chose this simple methodology, more adequate for our study in Amazon, regarding the lack of infrastructure in the field work in the Amazon region. However, it has some limitations. The literature cites several studies that analyzed this biomarker, showing an increase or decrease of the levels of GSH in situations of exposure to Hg (Grotto et al., 2010; Rangel-Méndez et al., 2016; Stacchiotti et al., 2009). These studies corroborate our findings showing evidence of exposure to mercury and GSH (Figure 4A).

The studies of Farina et al., 2013 and Patrick, 2002, were important references of our manuscript. In these articles, the relationship between exposure to MeHg and GSH was deeply analyzed in metabolic terms, and these texts were great sources of information for the discussion of our results.

The studies of Farina et al., 2013 and Patrick, 2002, were important references of our manuscript. In these articles, the relationship between exposure to MeHg and GSH was deeply analyzed in metabolic terms.

One consideration that can be accepted as the control of the sample is that all samples were stored at the same time in ultrafreezer (-80 ºC). This corroborates that if oxidation occurred, it was the same for all samples. This can be seen when comparing the GSH between the groups, if the oxidation with loss of GHS occurred, it was homogeneous for all the samples. Therefore, the means and medians of this biomarker can be compared among the study communities.

We expected differences between the levels of GSH among the communities studied, but this result was not observed. This subject was discussed in the text (page 11, lines 332-340). In the manuscript, we discuss the fact that the observed GSH results may be due to high selenium (Se) consumption, present in Brazilian nuts, which have high dietary antioxidant levels. Thus, maintaining stable GSH levels (non-depleted) in the Hg-exposed populations. This possibility is presented but we did not deepen the discussion on these issues, as they are the subject of another article previously published by our research group, Vega et al., 2017. In the aforementioned study, the authors explore the issues related to fish consumption, nut consumption and Se levels, which may be the main sources of Se for the Cuniã community.

We chose to insert in the discussion the following sentence, showing the fragility of the analysis of this biomarker: "Another important factor to consider, that may have contributed to find no differences of GSH among the studied communities, is that this molecule undergoes self- oxidation very easily, depending on the conditions of collection and storage (Giustarini et al., 2016). The GSH auto-oxidation of our samples, due to the difficulties of field work in a remote region, may have interfered in the results obtained, leading to this indifference of GSH levels between the groups mentioned.”

Cited references:

Vega, C.M.; Godoy, J.M.; Barrocas, P.R.G.; Gonçalves, R.A.; Oliveira, B.F.A.; Jacobson, L.V.; Mourão, D.S.; Hacon, S.S. Selenium levels in the whole blood of children and teenagers from two riparian communities at the Madeira river basin in the Western Brazilian Amazon. Biol. Trace Elem. Res., 2017, 175 (1), 87–97.

Giustarini, D.; Tsikas, D.; Colombo, G.; Milzani, A.; Dalle-Donne, I.; Fanti, P.; Rossi, R. Pitfalls in the analysis of the physiological antioxidant glutathione (GSH) and its disulfide (GSSG) in biological samples: An elephant in the room. Journal of chromatography. B, Analytical technologies in the biomedical and life sciences, 2016, 1019 (15), 21–28.

Grotto, D.; Valentini, J.; Fillion, M.; Passos, C. J. S.; Garcia, S. C.; Mergler, D.; Barbosa, F. Mercury Exposure and Oxidative Stress in Communities of the Brazilian Amazon. Sci. Total Environ., 2010, 408 (4), 806–811.

Rangel-Méndez, J. A.; Arcega-Cabrera, F. E.; Fargher, L. F.; Moo-Puc, R. E. Mercury Levels Assessment and Its Relationship with Oxidative Stress Biomarkers in Children from Three Localities in Yucatan, Mexico. Sci. Total Environ., 2016, 543 (Pt A), 187–196.

Stacchiotti, A.; Morandini, F.; Bettoni, F.; Schena, I.; Lavazza, A.; Grigolato, P. G.; Apostoli, P.; Rezzani, R.; Aleo, M. F. Stress Proteins and Oxidative Damage in a Renal Derived Cell Line Exposed to Inorganic Mercury and Lead. Toxicology, 2009, 264 (3), 215–224.

Farina, M.; Avila, D. S.; da Rocha, J. B. T.; Aschner, M. Metals, Oxidative Stress and Neurodegeneration: A Focus on Iron, Manganese and Mercury. Neurochem. Int., 2013, 62 (5), 575–594.

Patrick, L. Mercury Toxicity and Antioxidants: Part 1: Role of Glutathione and Alpha-Lipoic Acid in the Treatment of Mercury Toxicity. Altern. Med. Rev., 2002, 7 (6), 456–471.

Reviewer 2 Report

The paper "oxidative stress level induced by mercury exposure in Amazon juvenile population in Brazil" provides interesting data that is likely to be of interest to other investigators. However, a number of concerns that require further attention are below indicated:

Page 2 , line 50:  "aN organic form" . N should be put in lower case

page 5  line 143:  The measurement of total GSH cannot give information about oxidative state in plasma.The plasma/serum get fast oxidised as soon as it is collected ( here the samples were frozen before analysis). Moreove GSH also exchange with other thiols, low and high molecular weight , therefore if the authors wants to use GSh as a potential biomarker the have to do the analysis on the fresh sample and immediatly after collection treat the samples for reduce, oxidised and GSH bind to proteins concentration. This measurement will be indicative of a di-sbalance in the redox homeostasis in plasma. The authors have to provide ana explanation well refenced and solid to explain why they expected a change in total GSH content.

Page 7 , line 227: The author did not detecte differences in the total GSH and this is the correct value expected because the liver would compensate for changes in the  GSH level in the blood. As mentioned in the previous point, the interesting data should be the ratio between GSH/GSSG and GSH/PSSG. As suggestion the authors could also measure the ratio CSH/CSSC because CSH is the main thiol present in plasma/serum , therefore the first antioxidant molecule that would be affected in case of oxidative stress.

Page 8 , Figure 3: Even if the graphical representation is useful to understand better the data, figure 3 can be put as supplemental material since the data are already presented in table 1 and the Figure 3 is a repetition of what already shown . Moreover, it is not clearly indicated what the small dots reported outside the box plot are.

Page 10, line 295: the authors mentioned "table 3 but is not included in the manuscript.

              line 301: "GSH grouping " is not part of the established nomenclature to express "increasing concentration of GSH" . Same comment for "selenol grouping" page 11 line 325

              line 308 : the sentence can be rephrase to have higher clarity.  "MDA concentration did not increase significantly because in the range of the redox buffer capacity ...." is this the meaning of the sentence?

              line 319: the main glutathione form in serum is GSSG and as explained before the authors should not expect a change in total amount of glutathone but a change in the ration of the different species. a complete profile of thiols (CSH , homocysteine , CSSC, GSH ,GSSG and thiolated proteins) would most likly give the results expected by the authors.

Author Response

Dear Reviewer

Thank you for the opportunity to review our manuscript. We very much appreciate the comments and suggestions from your revision, and maked the necessary adjustments in the manuscript.

All recommendations and questions were answered below (in red).

Thank you again for your time and consideration.

In case of any questions, please do not hesitate to contact us anytime.

Best wishes,

Leandro Vargas B. de Carvalho

Response to Reviewer 2 Comments

Moderate English changes required

Response: The manuscript was entirely revised by an english native speaker.

The paper "oxidative stress level induced by mercury exposure in Amazon juvenile population in Brazil" provides interesting data that is likely to be of interest to other investigators. However, a number of concerns that require further attention are below indicated:

1) Page 2, line 50:  "aN organic form". N should be put in lower case

1) R. The change was made in the revised text.

2) Page 5 line 143:  The measurement of total GSH cannot give information about oxidative state in plasma. The plasma/serum get fast oxidized as soon as it is collected (here the samples were frozen before analysis). Moreover, GSH also exchange with other thiols, low and high molecular weight, therefore if the authors wants to use GSH as a potential biomarker the have to do the analysis on the fresh sample and immediately after collection treat the samples for reduce, oxidized and GSH bind to proteins concentration. This measurement will be indicative of a disbalance in the redox homeostasis in plasma. The authors have to provide an explanation well refenced and solid to explain why they expected a change in total GSH content.

2) R. Our method employed in the GSH analysis did not use any feature to stabilize the serum. However, the samples were storage in ultrafreezer (-80 ºC), to control, reduce and/or avoid the oxidation of the GSH.

Serum GSH analysis was one of several biomarkers to evaluate the levels of oxidative stress induced by Hg uptake via fish consumption.

For GSH analysis the classical method (Hu, 1994) was used, by adding the DTNB Ellman's reagent. Analyzes plasma/serum GSH, as well as all other plasma/serum thiol groups, also were included in our study. So, we chose this simple methodology, more adequate for our study in Amazon, regarding the lack of infrastructure in the field work in the Amazon region. However, it has some limitations. The literature cites several studies that analyzed this biomarker, showing an increase or decrease of the levels of GSH in situations of exposure to Hg (Grotto et al., 2010; Rangel-Méndez et al., 2016; Stacchiotti et al., 2009). These studies corroborate our findings showing evidence of exposure to mercury and GSH (Figure 4A).

The studies of Farina et al., 2013 and Patrick, 2002, were important references of our manuscript. In these articles, the relationship between exposure to MeHg and GSH was deeply analyzed in metabolic terms, and these texts were great sources of information for the discussion of our results.

The studies of Farina et al., 2013 and Patrick, 2002, were important references of our manuscript. In these articles, the relationship between exposure to MeHg and GSH was deeply analyzed in metabolic terms.

One consideration that can be accepted as the control of the sample is that all samples were stored at the same time in ultrafreezer (-80 ºC). This corroborates that if oxidation occurred, it was the same for all samples. This can be seen when comparing the GSH between the groups, if the oxidation with loss of GHS occurred, it was homogeneous for all the samples. Therefore, the means and medians of this biomarker can be compared among the study communities.

We expected differences between the levels of GSH among the communities studied, but this result was not observed. This subject was discussed in the text (page 11, lines 332-340). In the manuscript, we discuss the fact that the observed GSH results may be due to high selenium (Se) consumption, present in Brazilian nuts, which have high dietary antioxidant levels. Thus, maintaining stable GSH levels (non-depleted) in the Hg-exposed populations. This possibility is presented but we did not deepen the discussion on these issues, as they are the subject of another article previously published by our research group, Vega et al., 2017. In the aforementioned study, the authors explore the issues related to fish consumption, nut consumption and Se levels, which may be the main sources of Se for the Cuniã community.

We chose to insert in the discussion the following sentence, showing the fragility of the analysis of this biomarker: "Another important factor to consider, that may have contributed to find no differences of GSH among the studied communities, is that this molecule undergoes self- oxidation very easily, depending on the conditions of collection and storage (Giustarini et al., 2016). The GSH auto-oxidation of our samples, due to the difficulties of field work in a remote region, may have interfered in the results obtained, leading to this indifference of GSH levels between the groups mentioned.”

Cited references:

Vega, C.M.; Godoy, J.M.; Barrocas, P.R.G.; Gonçalves, R.A.; Oliveira, B.F.A.; Jacobson, L.V.; Mourão, D.S.; Hacon, S.S. Selenium levels in the whole blood of children and teenagers from two riparian communities at the Madeira river basin in the Western Brazilian Amazon. Biol. Trace Elem. Res., 2017, 175 (1), 87–97.

Giustarini, D.; Tsikas, D.; Colombo, G.; Milzani, A.; Dalle-Donne, I.; Fanti, P.; Rossi, R. Pitfalls in the analysis of the physiological antioxidant glutathione (GSH) and its disulfide (GSSG) in biological samples: An elephant in the room. Journal of chromatography. B, Analytical technologies in the biomedical and life sciences, 2016, 1019 (15), 21–28.

Grotto, D.; Valentini, J.; Fillion, M.; Passos, C. J. S.; Garcia, S. C.; Mergler, D.; Barbosa, F. Mercury Exposure and Oxidative Stress in Communities of the Brazilian Amazon. Sci. Total Environ., 2010, 408 (4), 806–811.

Rangel-Méndez, J. A.; Arcega-Cabrera, F. E.; Fargher, L. F.; Moo-Puc, R. E. Mercury Levels Assessment and Its Relationship with Oxidative Stress Biomarkers in Children from Three Localities in Yucatan, Mexico. Sci. Total Environ., 2016, 543 (Pt A), 187–196.

Stacchiotti, A.; Morandini, F.; Bettoni, F.; Schena, I.; Lavazza, A.; Grigolato, P. G.; Apostoli, P.; Rezzani, R.; Aleo, M. F. Stress Proteins and Oxidative Damage in a Renal Derived Cell Line Exposed to Inorganic Mercury and Lead. Toxicology, 2009, 264 (3), 215–224.

Farina, M.; Avila, D. S.; da Rocha, J. B. T.; Aschner, M. Metals, Oxidative Stress and Neurodegeneration: A Focus on Iron, Manganese and Mercury. Neurochem. Int., 2013, 62 (5), 575–594.

Patrick, L. Mercury Toxicity and Antioxidants: Part 1: Role of Glutathione and Alpha-Lipoic Acid in the Treatment of Mercury Toxicity. Altern. Med. Rev., 2002, 7 (6), 456–471.

3) Page 7, line 227: The author did not detected differences in the total GSH and this is the correct value expected because the liver would compensate for changes in the GSH level in the blood. As mentioned in the previous point, the interesting data should be the ratio between GSH/GSSG and GSH/PSSG. As suggestion the authors could also measure the ratio CSH/CSSC because CSH is the main thiol present in plasma/serum, therefore the first antioxidant molecule that would be affected in case of oxidative stress.

3)R. Thank you for the comment, we appreciate the information and agree with it.  Since, is important to interpret the GSH results.

At the time of this study we did not perform the analysis of the GSH/GSSH relationship, because at the time of this analysis of the biomarkers, it was not possible to develop this method (GSH/GSSG ratio), therefore only GSH was determined.

4) Page 8, Figure 3: Even if the graphical representation is useful to understand better the data, figure 3 can be put as supplemental material since the data are already presented in table 1 and the Figure 3 is a repetition of what already shown. Moreover, it is not clearly indicated what the small dots reported outside the box plot are.

4) R. We believe that this figure is an adequate visual representation of the differences found between the analyzed biomarkers within each population studied, and also shows how the biomarkers values increase among those who consume fish more frequently. Although these data are present in Table 1, Figure 3 presents this information with a much greater clarity, giving more emphasis to the results, and highlighting the data found in the statistical analyzes performed. We request that this figure be maintained in this part of the manuscript because the visual information brings value to graphically present the results obtained in our study. Small dots outside the box plot represents the outliers, and It was insert in the text of manuscript.

5) Page 10, line 295: the authors mentioned "table 3 but is not included in the manuscript.

5) R. Correction was made in the manuscript. The correct number of the Table cited in this part is Table 2

6) Page 10, line 301: "GSH grouping " is not part of the established nomenclature to express "increasing concentration of GSH". Same comment for "selenol grouping" page 11 line 325.

6) R. The suggestion was accepted, and the changes were made to the revised text. The term "grouping" was deleted.

7) Page 10, line 308: the sentence can be rephrased to have higher clarity.  "MDA concentration did not increase significantly because in the range of the redox buffer capacity ...." is this the meaning of the sentence?

7) R. The suggestion was accepted, and the changes were made to the revised text. The sentence was edited to “MDA concentrations do not increase significantly, probably because within this range of Hg concentration, the redox equilibrium is maintained without generating oxidative damage”.

8) Page 10, line 319: the main glutathione form in serum is GSSG and as explained before the authors should not expect a change in total amount of glutathione but a change in the ration of the different species. a complete profile of thiols (CSH, homocysteine, CSSC, GSH, GSSG and thiolated proteins) would most likely give the results expected by the authors.

8) R. Unfortunately, it was not possible to carry out other methodologies for all these biomarkers. Of course, these analyzes of other biomarkers would greatly enrich our results.